# Safety of Oral *Carica papaya* L. Leaf 10% Ethanolic Extract for Acute and Chronic Toxicity Tests in Sprague Dawley Rats

**DOI:** 10.3390/toxics12030198

**Published:** 2024-03-01

**Authors:** Weerakit Taychaworaditsakul, Chalermpong Saenjum, Nongkran Lumjuan, Kriangkrai Chawansuntati, Suphunwadee Sawong, Kanjana Jaijoy, Mingkwan Na Takuathung, Seewaboon Sireeratawong

**Affiliations:** 1Clinical Research Center for Food and Herbal Product Trials and Development (CR-FAH), Faculty of Medicine, Chiang Mai University, Chiang Mai 50200, Thailand or weerakit.tay@gmail.com (W.T.); suphunwadee.sa@cmu.ac.th (S.S.); mingkwan.n@cmu.ac.th (M.N.T.); 2Department of Pharmacology, Faculty of Medicine, Chiang Mai University, Chiang Mai 50200, Thailand; 3Department of Pharmaceutical Science, Faculty of Pharmacy, Chiang Mai University, Chiang Mai 50200, Thailand; chalermpong.saenjum@gmail.com; 4Research Institute for Health Sciences, Chiang Mai University, Chiang Mai 50200, Thailand; nongkran.l@cmu.ac.th (N.L.); kriangkrai.ch@cmu.ac.th (K.C.); 5McCormick Faculty of Nursing, Payap University, Chiang Mai 50000, Thailand; joi.kanjana@gmail.com

**Keywords:** *Carica papaya* L. leaf extract, medicinal plants, acute toxicity, chronic toxicity, safety evaluations, animal

## Abstract

*Carica papaya L.* leaves, traditionally utilized in dietary supplements and pharmaceuticals, exhibit a broad spectrum of potentially therapeutic properties, including anti-inflammatory, antimalarial, and wound healing properties. This study examined the acute and chronic toxicity of 10% ethanolic-extracted *C. papaya* leaf in Sprague Dawley rats. The acute toxicity assessment was a single oral dose of 5000 mg/kg body weight, while the chronic toxicity assessment included daily oral doses of 100, 400, 1000, and 5000 mg/kg over 180 days. Systematic monitoring covered a range of physiological and behavioral parameters, including body and organ weights. End-point evaluations encompassed hematological and biochemical analyses, along with gross and histopathological examinations of internal organs. Findings revealed no acute toxicity in the *C. papaya* leaf extract group, although a significant decrease in uterine weight was observed without accompanying histopathology abnormalities. In the chronic toxicity assessment, no statistically significant differences between the control and the *C. papaya* leaf extract groups were detected across multiple measures, including behavioral, physiological, and hematological indices. Importantly, histopathological examination corroborated the absence of any tissue abnormalities. The study results indicate that *C. papaya* leaf extract exhibited no adverse effects on the rats during the 180-day oral administration period, affirming its potential safety for prolonged usage.

## 1. Introduction

The *Carica papaya* L. plant, belonging to the *Caricaseae* family, is indigenous to Central America and the southern regions of Mexico, but has been widely cultivated in many Asian countries, from the Philippines to Pakistan. This plant is recognized globally for its extensive therapeutic applications which have been delineated in a variety of scientific investigations [1]. The nutritional value of the fruit of *C*. *papaya*, known as papaya, is well known globally, yet only minimal attention has been accorded to its leaves, seeds, and roots, which also offer medicinal properties. Previous comprehensive phytochemical investigation has demonstrated that *C. papaya* leaf extract is rich in bioactive components such as flavonoids, alkaloids, saponins, phytosterols, phenolics, and tannins, highlighting its prospective pharmaceutical utility [2,3,4,5]. Although existing research has predominantly addressed the extract’s effectiveness in treating dengue fever [4], it has also been shown to function as an anti-inflammatory agent, selectively modulating cytokine production [6,7]. Additionally, empirical evidence underscores its multifaceted therapeutic benefits, including anti-bacterial [8], anti-malarial [9], gastroprotective [10], hypoglycemic [11], hypolipidemic [12], anti-oxidant [13], hematological [14], anti-tumor activities [3] and wound-healing capacities [15]. Various extraction methods, ranging from aqueous to ethanol-based processes, have been optimized to facilitate its application across an array of medical conditions [5,16,17]. The existing body of research accentuates the leaves’ prospective utility as a formalized herbal medicine product, indicating not only its potential for disease prevention and health promotion, but also its significant economic value.

Ensuring the safety and proper formulation of medicinal herbs is as critical as establishing their efficacy. As outlined in the World Health Organization (WHO) 2019 Global Report on Traditional and Complementary Medicine, herbal products often undergo stringent safety evaluations similar to those mandated for conventional pharmaceuticals, including post-marketing surveillance. Regulatory standards for traditional herbal medicine frequently incorporate contemporary scientific research on comparable market products [18]. In line with the Principles of Good Laboratory Practice (GLP) of the Organization for Economic Co-operation and Development (OECD), government regulatory agencies often mandate both general and specific animal toxicity studies of varying durations before issuing an herbal medicine registration [18]. Despite the demonstrated effectiveness of *C. papaya* extracts, questions persist concerning their long-term safety. Acute in vivo toxicity studies have reported that rats administered freeze-dried aqueous extract of *C. papaya* leaf at dosages ranging from 5 to 2000 mg/kg body weight (BW) exhibit no discernable adverse effects [19]. Short-term studies (up to thirteen weeks) involving daily doses of 2000 mg/kg have reported neither morbidity nor mortality in rats. Data on long-term health implications for chronic users, however, remain conspicuously absent [20].

Toxicological evaluations serve as a cornerstone in pharmacological research and pharmaceutical development, helping to establish a compound’s safety profile before its clinical application in humans [21,22]. Given this imperative, chronic toxicity studies complement acute evaluations to provide a comprehensive assessment of an herbal medicine’s long-term physiological, biochemical, hematological, and pathological impact. This study aims to assess both the acute and chronic toxicity profiles of 10% ethanolic-extracted *C. papaya* leaf, utilizing Sprague Dawley rats as the experimental model.

## 2. Materials and Methods

### 2.1. Chemicals and Reagents

Absolute ethanol, acetic acid, acetonitrile, gallic acid, rutin, quercetin, rosmarinic acid, luteolin, apigenin, and kaempferol were purchased from Sigma-Aldrich Chemical Company (St. Louis, MO, USA). All other chemicals were of analytical grade.

### 2.2. Plant Material

This study employed naturally cultivated *C. papaya* (Holland papaya) from Mae On District, Chiang Mai, Thailand. Leaves of *C. papaya* were harvested from 3- to 4-month-old plants between January and July 2021. The fresh leaves served as the source material for this investigation. Expert botanists authenticated the plant specimens using taxonomic photographs and reference samples deposited in the Plant Museum of the Faculty of Pharmacy at Chiang Mai University (voucher specimen number 0023307). Subsequently, the extracted *C. papaya* leaves were analyzed using Thai Herbal Pharmacopoeia methods [23], supervised by Associate Professor Dr. Chalermpong Saenjum from the Faculty of Pharmacy at Chiang Mai University.

### 2.3. Extract Preparation

The leaves of *C. papaya* were initially cleaned and depurated of their central stems and large branches, followed by triple rinsing with distilled water. The cleansed leaves were then subjected to hot air oven drying at 50 °C for 12 h. Subsequently, the dried material was pulverized using an 80-mesh sieve grinder (Nanotech NT-1000D; Thanapan Ltd., Chiang Mai, Thailand) to achieve a homogenous particle size of less than 177 microns, maximizing the surface area for optimal extraction efficiency. This processed leaf powder was securely stored in a vacuum-sealed bag protected from light and stored at -20 °C, and dry environment until further use. For the extraction process, 8 kg of *C. papaya* leaf powder were prepared in 60 L of 10% ethanol in water, using high-speed extraction (EURO Best Technology Co., Ltd., Bangkok, Thailand) at 150 rpm and a temperature of 80–90 °C for 3 h. The obtained solution was filtered and then centrifuged to separate the residue, after which extracted twice with 10% ethanol. The separated solution was then weighed, adjusted to a total of 150 kg using 10% *v*/*v* ethanol in water solution. The concentration of maltodextrin was 0.5% *w*/*v* and homogenized at 5000 rpm. Then, the obtained solution was spray dry (SDE-50 EURO Best Technology Co., Ltd., Bangkok, Thailand), with an inlet temperature of 160 °C, and the flow rate was set at 500 mL/min to yield a granular, water-soluble powder with a 2.81% extraction efficiency.

### 2.4. Phenolic and Flavonoid Content Identification Via High-Performance Liquid Chromatography (HPLC)

The *C. papaya* leaf extract was subjected to HPLC for quantification and the identification of various phenolic and flavonoid compounds, including, but not limited to, gallic acid, rutin, quercetin, rosmarinic acid, luteolin, apigenin, and kaempferol, under specific analytical conditions following Phromnoi et al. [24]. In brief, chromatographic separation was achieved using a Symmetry-Shield^®^ RP18 column (250 mm × 4.6 mm ID (internal diameter)) supplied by Waters Corporation (Waters Co., Ltd., Milford, MA, USA). The mobile phases were formulated from 0.1% acetic acid in acetonitrile and deionized water mixed in a 30:70 ratio, under isocratic conditions at a flow rate of 1.0 mL/min. Each 10 μL sample was introduced into the column and monitored through a diode array detector at wavelengths of 278 and 325 nm. The amounts of each detected compound in the *C. papaya* extract were calculated and expressed as mg/g extract.

### 2.5. The Animal Subjects and Ethical Considerations

This extract was preserved in opaque, airtight containers under cool and dry conditions. Sprague Dawley rats of both sexes, weighing between 180 and 200 g, were employed in this study. The animals were bred by the National Laboratory Animal Center at Mahidol University, Nakhon Pathom, Thailand, and subsequently transferred to the laboratory facilities for acclimatization. Ambient conditions were controlled at a temperature of 25 ± 1 °C and a relative humidity of 60%, with a 12 h light-dark cycle. The rats were provided with ad libitum access to food and water. To facilitate animal welfare, each rat was allowed a minimum acclimatization period of one week prior to the start of the experiment. This study was conducted in compliance with ethical guidelines and received approval from the Research Ethics Committee for Animal Studies of the Faculty of Medicine at Chiang Mai University, Thailand (approval code: 7/2564).

### 2.6. Hippocratic Assessment

Building on established methods used in previous research [25], this study employed Hippocratic screening to evaluate the safety profile of *C. papaya* leaf extract. Specifically, two female rats were orally administered a single dose of 2000 mg/kg and observed individually in open fields at 5, 10, 15, 30, 60, 120, and 240 min after administration and 24 h a day for clinical signs and mortality, as measured by motor activity (observing animal movements using infrared (I.R.) beams in a motor activity cage), respiration rate (observing the breathing of an animal in one minute), righting reflex (considering the behavior of an animal that turns and flips over), and screen grip (considering the ability of an animal to cling onto a cage using its forelimbs and hindlimbs). The assessment aimed to identify any potential adverse reactions, e.g., sedation, emesis, muscular spasms, and watery diarrhea.

### 2.7. Acute Toxicity Evaluation of C. papaya Leaf Extract

In accordance with WHO and OECD Test Guideline 420 [26,27], the rats were randomly assigned to either a treatment group (*n* = 5) which was administered a single oral dose of 5000 mg/kg of *C. papaya* leaf extract, or to a control group (*n* = 5), receiving 2 mL/kg of distilled water by oral gavage. Acute toxicity indicators, e.g., lethargy, emesis, muscle spasms, and watery diarrhea, were initially monitored for a six-hour period and then subsequently on a daily basis for a 14-day period. Additionally, body weight was recorded on days 7 and 14 as were any mortality incidents. At the conclusion of the observation period, the rats were euthanized using intraperitoneal thiopental sodium injection at a dose of 120 mg/kg, after which the animals underwent a physical examination by us checking their vital signs, pulse, and reflexes to confirm death and a comprehensive gross examination of internal organs, including but not limited to the lungs, heart, liver, and kidneys, was performed. All harvested organs were subsequently preserved in 10% formaldehyde for further analysis.

### 2.8. Assessment of Chronic Toxicity of C. papaya Leaf Extract

The chronic toxicity evaluation was performed in accordance with WHO and OECD Test Guideline 452 [27,28]. A total of 100 rats, half male and half female, were divided into seven groups. Group 1 (*n* = 10), which served as the control group, received daily doses of 2 mL/kg of distilled water for 14 days. Group 2 (*n* = 10), designated as the satellite control group, also received 2 mL/kg of distilled water daily for the same period and was observed for an additional 28 days post-treatment (*n* = 5). Groups 3 to 6 (*n* = 10 per each) were the treatment groups and were administered doses of 100, 400, 1000, and 5000 mg/kg, respectively, of *C. papaya* leaf extracts for 180 days. Group 7 (*n* = 10) functioned as a satellite control, receiving 5000 mg/kg of *C. papaya* leaf extract daily for 14 days and was observed for an additional 28 days post-treatment (*n* = 5). All doses were administered by oral gavage.

During the observation periods, behavioral patterns, clinical indicators, and body weights were systematically monitored and documented. Any deceased animals were immediately subjected to a necropsy. On day 180, hematological and biochemical parameters were analyzed from collected blood samples, after which all rats were euthanized with a 120 mg/kg intraperitoneal thiopental sodium injection, and their vital signs, pulse, and reflexes were checked to confirm death and underwent comprehensive macroscopic and microscopic evaluation of a range of internal organs, including the lungs, heart, liver, pancreas, kidneys, stomach, intestines, spleen, adrenal glands, ovaries, uterus, testes, eyes, brain, muscles, and nerves.

### 2.9. Statistical Analysis Methodology

The results are presented as mean ± S.E.M. For evaluating the data obtained from the acute toxicity study, either a t-test or the Mann–Whitney U test was employed, as appropriate. For chronic toxicity tests, the data underwent an initial analysis using the Shapiro–Wilk test to assess normality. In cases where the Shapiro–Wilk test showed no significant deviation from a normal distribution, an analysis of variance (ANOVA) was conducted, followed by Tukey’s multiple comparison tests. Conversely, if the Shapiro–Wilk test indicated other than a normal distribution, the data were subjected to analysis using the Kruskal–Wallis nonparametric ANOVA test, followed by Dunn’s test. Statistical significance was set at *p*-values less than 0.05. The statistical analyses were performed using IBM SPSS Statistics, version 22.0 (International Business Machines Corporation, Armonk, NY, USA).

## 3. Results

### 3.1. Quality Control and Standard Specification for C. papaya Leaf Extract

Table 1 presents the physical characteristics of desiccated *C. papaya* leaves, analyzed using randomly selected samples. An analysis of the six parameters presented in Table 1, including acid-insoluble ash, loss on drying, and total ash, provides a description of the qualities and characteristics of the raw materials used for the preparation of the extracts for this experiment.

Remarkably, the *C. papaya* leaf extract met all regulatory requirements set forth by the Food and Drug Administration (FDA) for food additives. Specifically, the extract was within permissible limits for heavy metals including arsenic, mercury, lead, and cadmium. Additionally, the extract was determined to be free from contamination by key pathogenic microorganisms, including *E. coli*, *S. aureus*, *Clostridium* spp., and *Salmonella* spp., as shown in Table 2. Additionally, comprehensive pesticide screening revealed no detectable residues of any of the 20 organochlorines, 8 organophosphates, 6 pyrethroids, and 9 carbamates tested, as shown in Appendix A. These stringent quality control standards affirm the extract’s safety and suitability for therapeutic application as well as demonstrating the value of further investigation.

In this study, the phenolic and flavonoid constituents of *C. papaya* leaf extract were comprehensively characterized using HPLC, coupled with ultraviolet (UV) and mass spectrometry detectors. As shown in Figure 1 and Table 3, the extract contained key bioactive compounds, including rutin, quercetin, gallic acid, catechin, apigenin, and kaempferol.

### 3.2. Preliminary Toxicity Assessment of C. papaya Extract in Female Rats

In accordance with the Hippocratic screening protocol, a preliminary evaluation was conducted to assess the potential toxicity of oral administration of *C. papaya* leaf extract at a dosage of 2000 mg/kg in female rats. Observations were carried out over a 24 h period and are detailed in Table 4. No adverse behavioral changes or fatalities were observed within this timeframe. Additionally, gross examination of the visceral organs and overall carcass characteristics yield no detachable abnormalities.

### 3.3. Acute Toxicity Assessment of C. papaya Leaf Extract in Female Rats

An acute evaluation was performed utilizing a single oral dose of 5000 mg/kg of *C. papaya* leaf extract administered to female rats. During the initial 24 h observation period, no behavioral aberrations were detected in any of the experimental groups or in the control groups. No fatalities were recorded in any of the groups during the study. Table 5 presents body weight measurements on days 7 and 14, revealing no statistically significant divergence from the control group (*p* < 0.05).

No mortality ensued following administration of a single dose of 5000 mg/kg of the extract, and normal behavioral patterns remained consistent. No pathological changes were noted across the range of physiological parameters, including ocular, dermal, fur, mucosal, respiratory, and circulatory assessments, as well as the autonomic and central nervous systems. At day 14, both body and internal organ weights of the experimental group remained statistically comparable to the control group (Table 6). Notably, although a significant decrease in uterine weight was observed, gross and histopathological examinations confirmed the absence of any discernible abnormalities in uterine size, color or texture (Figure 2).

### 3.4. Assessment of Chronic Toxicity Induced by C. papaya Leaf Extract

#### 3.4.1. Variation in Body and Organ Weight

Table 7 and Table 8 present longitudinal body weight metrics for female and male rats subjected to a six-month regimen of *C. papaya* leaf extract administration. On day 90, a rigorous statistical analysis unveiled a notable and statistically significant decrease in the body weight of female rats in the low-dose group (100 mg/kg) as compared to the control group. Similarly, on day 30, a significant reduction in the body weight of male rats was observed in the high-dose satellite group (5000 mg/kg) in comparison to the control group. Conversely, Table 9 and Table 10 depict that there were no discernible changes in the internal organ weights of either female or male rats.

#### 3.4.2. Hematological and Biochemical Assessments

Hematological parameters were rigorously evaluated in rats following oral administration of *C. papaya* leaf extract as detailed in Table 11 and Table 12. Furthermore, Table 13 and Table 14 show the results of biochemical evaluations. Notably, both sets of evaluations revealed an absence of significant deviations from control values in rats subjected to chronic exposure at doses of 400, 1000, and 5000 mg/kg body weight per day. An exception to this uniformity was a statistically significant decline in alkaline phosphatase (ALP) levels, which was observed exclusively in male rats belonging to the satellite treatment group.

#### 3.4.3. Necropsy and Histopathological Evaluation in Chronic Toxicity Assessment

In response to the observed changes in ALP levels compared to the control group, comprehensive gross examinations were conducted to scrutinize organ appearance and characteristics. The outcomes affirm the absence of any macroscopic abnormalities. In addition, histopathological evaluations conducted on a range of vital organs of rats chronically administered *C. papaya* leaf extract corroborated these findings, indicating an absence of significant tissue injury (Figure 3 and Figure 4).

## 4. Discussion

The assessment of the safety of natural products is a critical prerequisite for obtaining regulatory approval, often necessitating a thorough evaluation of diverse scientific documents such as monographs, peer-reviewed articles, and scientific reports. International regulatory bodies like the European Food Safety Authority and the United States Food and Drug Administration place considerable emphasis on toxicological data derived from animal studies as a cornerstone for natural product safety evaluations [29,30]. While medicinal plants offer numerous health benefits, some can pose risks of adverse effects, including diseases and toxicity, necessitating rigorous scientific risk assessment investigations [31,32]. Systematic toxicity studies are pivotal for establishing safe consumption levels in humans, thereby helping to ensure the safe application of plant-derived products [33]. Although studies have explored the long-term toxicity of *C. papaya* leaf aqueous extracts in non-rodent animals such as rabbits [34], there remains a paucity of such investigations in rodent models. According to the ICH guideline M3(R2), a comprehensive safety profile should include toxicity studies in both rodent and non-rodent animal models [29]. While short-term studies have generally indicated a lack of toxic side effects associated with the use of *C. papaya* leaf extract [35,36], our study aimed to bridge the existing gap by focusing on its long-term chronic toxicity in rodents. This approach not only aligns with international guidelines but also provides a more robust scientific basis for the safety evaluation of this natural product. Nevertheless, the ethical considerations tied to animal testing in laboratories necessitate unwavering adherence to established protocols throughout the experimental procedure. Furthermore, it is imperative that both laboratory animal care and research activities are conducted within a standardized setting.

Preceding the commencement of clinical trials designed to assess the efficacy of herbal products, it is customary to initiate a preliminary phase involving acute and chronic toxicity tests in animal models. This foundational step is integral to determining optimal dosages and duration of administration, thereby establishing a solid groundwork for subsequent human trials [27]. In the current study, the dosages chosen were guided by existing human usage patterns for *C. papaya* leaf extract, which has previously been employed as a dietary supplement [37]. In humans, the daily dosage for this extract ranges from 1 to 50 g, which translates into a daily intake of 17 to 833 mg for an average 60 kg male [5,38,39]. To account for differences in metabolic rate among species, a body surface area exponent has been employed to standardize dosages [40]. The human dosages were then converted to rodent-appropriate dosages, resulting in the selection of a range of 103.5–5164.6 mg/kg body weight for our study. Ultimately, we opted for four dosage groups (100, 400, 1000, and 5000 mg/kg body weight) based on our research objectives. In addition, our unpublished data indicate that the range of effective doses of *C. papaya* leaf extract necessary to modulate platelet counts lies between 100 and 400 mg/kg body weight. For the chronic toxicity evaluation, the present study used doses of 100, 400, 1000, and 5000 mg/kg body weight, thereby encompassing both the effective and the high-dosage ranges to provide a comprehensive toxicity profile.

The quality control of herbal products is imperative for ensuring the standardization and safety of these natural remedies [41]. In accordance with the Thai Herbal Pharmacopoeia 2018, initial quality assessment methodologies for medicinal plant materials involves scrutinizing both the raw materials and the finished herbal products [23]. Good manufacturing practice (GMP) guidelines, as endorsed by the World Health Organization (WHO), advocate the utilization of high-performance liquid chromatography (HPLC) as an instrumental technique to quantify the phenolic and flavonoid content, thereby helping ensure product consistency [42]. In the present investigation, seven batches of *C. papaya* leaf extract all exhibited uniform concentrations of various bioactive compounds, including rutin, quercetin, gallic acid, catechin, apigenin, and kaempferol (Appendix A). These findings align with existing studies, which have identified an array of phytochemicals, e.g., flavonoids and phenolic compounds, saponins, cardiac glycosides, anthraquinones, and alkaloids [43,44]. Rutin, a predominant flavonoid in *C. papaya* leaves, has traditionally served as a chemical marker for quality control [38,45]. In addition to rutin, gallic acid has also been identified as another phenolic acid compound that offers a reliable indicator for product standardization [46]. Importantly, several of these compounds, particularly rutin, quercetin, and gallic acid, have been reported to exhibit potent anti-oxidant properties [47]. For that reason, our study employed these bioactive constituents, specifically phenolic acids like gallic acid and flavonoids such as rutin and quercetin, as quality control markers to ensure product consistency and efficacy.

The acute oral toxicity test serves as a critical evaluative method for assessing the immediate adverse effects following a substantial single oral dose of a substance within a 24 h timeframe. In this study, an elevated dose of 5000 mg/kg of *C. papaya* leaf extract was administered to assess its safety profile. This dosage was selected based on prior research utilizing a 2000 mg/kg dose for Hippocratic screening, a technique employed to uncover noteworthy pharmacological activities in medicinal plants, which yielded no toxic or lethal outcomes [25]. Our findings, presented in Table 6, substantiate that this high-dose administration of 10% ethanolic *C. papaya* leaf extract results in neither mortality nor significant body weight changes when compared to a control group. Moreover, comprehensive histological examination of various organs—including the brain, lungs, heart, liver, kidney, spleen, gastrointestinal tract, and reproductive organs—revealed no tissue damage. It is noteworthy, however, that the uterus in the high-dose group exhibited a significant decrease in size relative to the control, corroborating previous studies that have highlighted the uterotonic effects of *C. papaya* leaf extract [48,49]. Importantly, no mortality was observed in the present study, corroborating the existing literature that places the LD50 for *C. papaya* leaf extract well above 5000 mg/kg [50,51]. Additionally, histopathological analysis revealed no abnormalities in the uterine tissue, as indicated in Figure 2. These results underscore the need for rigorous evaluation of reproductive organs in chronic toxicity studies. Furthermore, given the uterotonic effects observed, caution may be warranted when considering the use of *C. papaya* leaf extract during pregnancy.

Chronic toxicity studies, typically lasting from six months to two years, are designed to assess the cumulative effects of prolonged exposure to a substance, particularly those adverse effects that may not become immediately apparent. This approach aligns with WHO guidelines which recommend that the duration of substance administration in animal models should mirror the anticipated duration of clinical exposure in humans [27]. A critical parameter for identifying potential toxicity is the assessment of changes in general behavior and body weight. The existing literature suggests that a loss of 10% or more of initial body weight in test animals could be a strong indicator of adverse side effects and may compromise survival [52,53,54]. However, our study revealed significant statistical fluctuations in rat weight. A decline was noted in female rats (low dose, 100 mg/kg) on day 90, and there was a notable reduction in the body weight of male rats (high dose, 5000 mg/kg) on day 30. Importantly, these observed changes, while perceptible, did not surpass a level of concern, specifically remaining below 10% weigh loss. Thus, our findings offer compelling evidence supporting the long-term safety of *C. papaya* leaf extract administration in a rodent model, thereby laying a foundation for potential translational applications in humans.

Fluctuations in the weight of internal organs are also widely acknowledged as sensitive markers for assessing the effects of drug exposure on organ integrity. Such assessments are integral to toxicological studies, where the organ weights of treated and untreated animals are systematically compared [55]. In the present investigation, gross pathological examinations revealed no significant abnormalities in the internal organs of *C. papaya* leaf extract-treated groups, i.e., the findings were consistent with those of the control groups. In accordance with the Organization for Economic Co-operation and Development (OECD) Test Guideline 452, the data garnered from this study substantiates the non-toxic profile of *C. papaya* leaf extract. Nevertheless, caution is advised in interpreting these results. A more in-depth evaluation of hematological, biochemical, and histopathological parameters, particularly in male satellite rats concerning spleen and epididymis, is warranted for a comprehensive safety assessment.

Blood chemistry and hematology analyses can serve as pivotal tools for detecting potential tissue damage or physiological stress, thereby providing valuable insights into the cellular integrity of internal organs. Given the circulatory system’s crucial role in disseminating nutrients and foreign substances throughout the organism, blood serves as a sensitive barometer of both physiological and pathological conditions [56,57]. Specifically, toxic substances can compromise key blood components such as red and white blood cells, platelets, and hemoglobin [58]. In the context of chronic toxicity, our study revealed no significant aberrations in blood parameters in either male or female rats following administration of *C. papaya* leaf extract. In the context of chronic toxicity, our study revealed no significant aberrations in blood parameters in either male or female rats following administration of *C. papaya* leaf extract and all values remained within clinically acceptable ranges [59,60,61,62].

To evaluate the potential impact on renal, hepatic, and pancreatic functions, comprehensive clinical blood chemistry assessments were conducted, focusing on key markers such as BUN, creatinine, AST, ALT, ALP, total protein, albumin, bilirubin, and glucose levels. Given the high volume of blood perfusion through the kidneys, this organ is particularly vulnerable to toxic insults. Substances deemed toxic are actively filtered by the kidneys, potentially leading to an accumulation in renal tubules [63], which underscores the relevance of BUN and creatinine as sensitive indicators of renal health [64]. In cases of renal, cortical and/or glomerular injury, an elevation in serum creatinine levels is typically observed [65]. Our study yielded no discernible toxic effects on either BUN or creatinine levels, supporting the assertion that *C. papaya* leaf extract does not adversely affect renal function. Corroborating these biochemical findings, histological examination of the kidneys from both treated and control groups revealed no discernible morphological differences, substantiating the absence of renal toxicity even after six months of chronic exposure to *C. papaya* leaf extract.

Liver function tests (LFTs) serve as a valuable diagnostic tool for identifying hepatic impairments. Bilirubin, a catabolic byproduct of hemoglobin, has a known association with hepatic diseases such as jaundice and primary biliary cirrhosis. Elevated bilirubin levels serve as a proxy for the severity of hepatic dysfunction [66]. In our investigation, markers indicative of hepatic synthetic capacity—specifically, bilirubin, total protein, and albumin—demonstrated no abnormal fluctuations [67]. In addition, AST and ALT, markers traditionally used to assess hepatic toxicity, showed no significant variations. These findings suggest that *C. papaya* leaf extract does not exert direct hepatotoxic effects or induce hepatotoxin-mediated liver injury [68,69]. Notably, a decline in ALP levels was observed, possibly indicating an adaptive or regenerative response of the liver to long-term exposure to the extract; however, these values remained within clinically accepted norms [59,60,61,62,70]. In addition, a satellite-treated group was employed to evaluate any latent or reversible toxic effects [71,72]. All assessed parameters in each of the groups remained within established reference ranges, affirming the extract’s safety profile. Nonetheless, to substantiate these biochemical indicators and further validate the hepatic safety of *C. papaya* leaf extract, supplemental histopathological assessments are recommended.

Generally, serum glucose levels exceeding 200 mg/dL in rats are indicative of “hyperglycemia” [73,74]. In our study, comparable plasma glucose levels were observed between the *C. papaya* leaf extract-treated and control groups. This finding suggests the absence of metabolic dysfunction or adverse side effects associated with prolonged use of the extract. The stable blood glucose levels in treated rats provides additional evidence of the absence of pancreatic islet damage.

To ascertain the safety profile of *C. papaya* leaf extract, comprehensive histopathological analyses were conducted on the internal organs of all test animals. Initial gross examination assessed qualitative attributes such as size, coloration, and overall appearance of various organs. These evaluations were followed by hematological tests and in-depth histological analyses. Our findings revealed no signs of toxicity or histopathological abnormalities across all treatment groups, particularly in organs of critical interest such as the liver, epididymis, and spleen in the male satellite group. The highest administered dose of 5000 mg/kg in rats translates to approximately 810 mg/kg of human body weight, according to established conversion factors [75]. This chronic toxicity evaluation substantiates the safety of *C. papaya* leaf extract for extended use.

## 5. Conclusions

In summary, the present study provides substantial evidence of the safety of *C. papaya* leaf extract. Acute toxicity tests using a single oral dose of 5000 mg/kg in rodents revealed no immediate adverse effects. However, notable changes in female reproductive organs, specifically the uterus, warrant further investigation. On the other hand, chronic administration of *C. papaya* leaf extract at dosages of 100, 400, 1000, and 5000 mg/kg/day over a 180-day period did not yield any observable toxicological effects. These findings strongly suggest that *C. papaya* leaf extract is amenable to long-term usage without inducing toxicity.

## Figures and Tables

**Figure 1 toxics-12-00198-f001:**
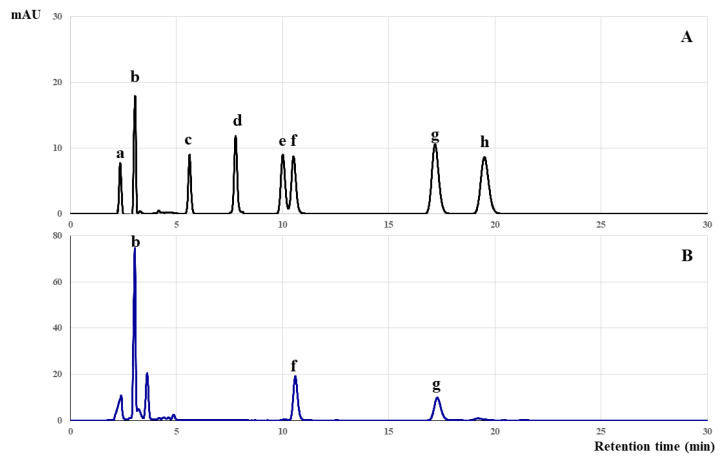
HPLC chromatograms of the reference standards (**A**) and the *C. papaya* leaf extract (**B**). The chromatograms indicate the quantities of key bioactive compounds identified, including gallic acid (a), rutin (b), rosmarinic acid (c), catechin (d), luteolin (e), quercetin (f), apigenin (g), and kaempferol (h).

**Figure 2 toxics-12-00198-f002:**
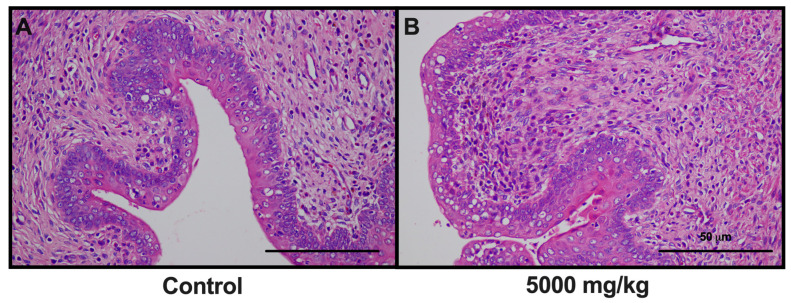
Histology of uterine tissue from the acute toxicity testing of *C. papaya* leaf extract in female rats (hematoxylin and eosin staining, 50×). Control (**A**); 5000 mg/kg BW (**B**).

**Figure 3 toxics-12-00198-f003:**
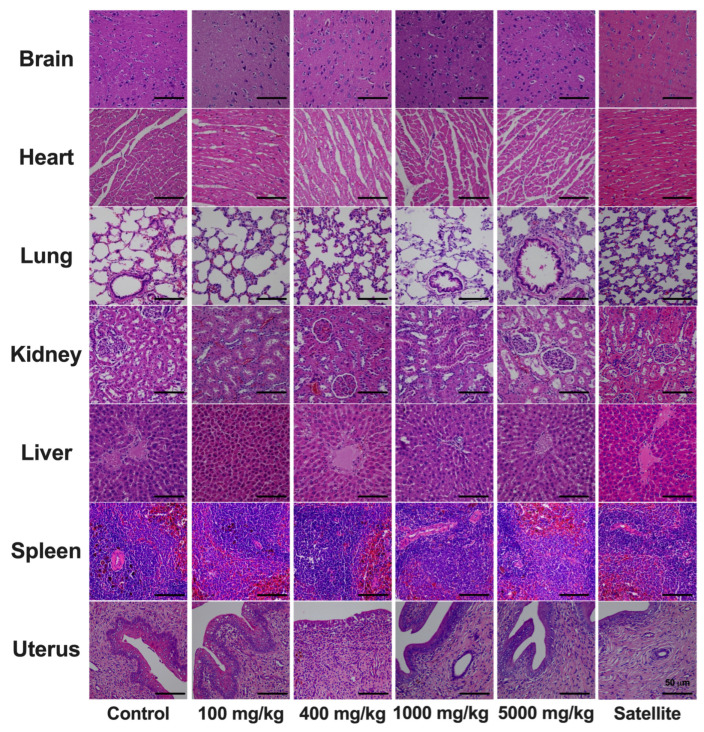
Histological sections of various tissues (brain, heart, lungs, kidneys, liver, spleen, and uterus) from female rats subjected to six-month chronic toxicity testing with *C. papaya* leaf extracts. All sections were stained with hematoxylin and eosin and imaged at 50× magnification.

**Figure 4 toxics-12-00198-f004:**
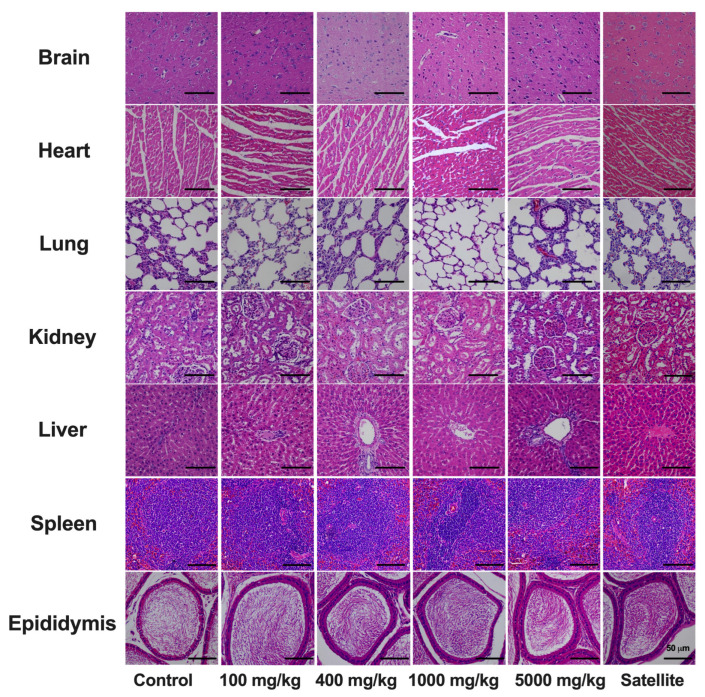
Histological sections of various tissues (brain, heart, lungs, kidneys, liver, spleen, and epididymis) from male rats subjected to six-month chronic toxicity testing with *C. papaya* leaf extracts. All sections were stained with hematoxylin and eosin and imaged at 50× magnification.

**Table 1 toxics-12-00198-t001:** Physical and chemical properties of *C. papaya* leaves.

Tests	Results
Foreign matter (%*w*/*w*)	Not found
Ethanol extractive content (%*w*/*w*)	16.56 ± 0.02
Chloroform-saturated aqueous extractive content (%*w*/*w*)	31.73 ± 0.05
Loss on drying (%*v*/*w*)	7.32 ± 0.01
Total ash (%*w*/*w*)	12.58 ± 0.01
Chemical composition	Phenolic acid, flavonoids

Values are expressed as mean ± S.E.M.

**Table 2 toxics-12-00198-t002:** Comparison of measured levels of heavy metal and microbe contamination and Food and Drug Administration (FDA) criteria.

Test Item	Results	FDA Criteria
Arsenic and heavy metal (mg/kg)		
Arsenic	0.356	Not more than 2.0
Mercury	0.015	Not more than 0.5
Lead	0.534	Not more than 1.0
Cadmium	<0.030	Not more than 0.3
Microbe contamination		
*Escherichia coli* (MPN/1 g)	Less than 3	Less than 3
*Clostidium* spp./0.1 g	Not found	Not found
*Staphylococcus aureus*/0.1 g	Not found	Not found
*Salmonella* spp./25 g	Not found	Not found

**Table 3 toxics-12-00198-t003:** Types and average quantities of phenolic and flavonoid compounds found in the analysis of production batches of *C. papaya* leaf extract.

Compound	Quantity Detected (mg/100 g Dry Extract)
Gallic acid	873.5 ± 4.9
Rutin	1043.0 ± 4.2
Catechin	421.5 ± 3.4
Rosmarinic acid	n.d.
Luteolin	n.d.
Quercetin	853.7 ± 3.7
Apigenin	105.6 ± 2.8
Kaempferol	18.56 ± 1.0

Values are expressed as mean ± S.E.M.; n.d.: not detectable.

**Table 4 toxics-12-00198-t004:** Hippocratic screening of *C. papaya* leaf extract in female rats.

Carica papaya L. Extract		Hours after Drug Administration																						
2000 mg/kg	1	2	3	4	5	6	7	8	9	10	11	12	13	14	15	16	17	18	19	20	21	22	23	24
Decrease in motor activity	0	0	0	0	0	0	0	0	0	0	0	0	0	0	0	0	0	0	0	0	0	0	0	0
Decrease in respiratory rate	0	0	0	0	0	0	0	0	0	0	0	0	0	0	0	0	0	0	0	0	0	0	0	0
Loss of righting reflex	0	0	0	0	0	0	0	0	0	0	0	0	0	0	0	0	0	0	0	0	0	0	0	0
Loss of screen grip	0	0	0	0	0	0	0	0	0	0	0	0	0	0	0	0	0	0	0	0	0	0	0	0
Time of death (hr.)	-	-	-	-	-	-	-	-	-	-	-	-	-	-	-	-	-	-	-	-	-	-	-	-

Decrease in motor activity: 0 = no decrease in motor activity, no change in respiratory rate, no loss of righting reflex, no loss of screen grip; +1 = does not move spontaneously, but when handled will move rapidly; +2 = when handled will move slowly; +3 = when handled will move sluggishly; +4 = when handled will not move at all. Decrease in respiration rate: +1 = 10% decrease in respiratory rate; +2 = 20% decrease in respiratory rate; +3 = 40% decrease in respiratory rate; +4 = 80% decrease in respiratory rate. Loss of righting reflex: +1 = can be placed only on one side; +2 = can be placed on either side equally well; +3 = can be placed on back as well as either side; +4 = cannot be aroused from back position by the hind leg toe pinch. Loss of screen grip: +1 = rat falls off at first shake of screen; +2 = rat falls off when screen has been inverted; +3 = rat falls off when the screen is at a 90° angle; +4 = rat falls off as the screen is tilted to a 45° angle.

**Table 5 toxics-12-00198-t005:** Body weight of female rats in the acute toxicity test.

Group	Body Weight (g)		
	Day 0	Day 7	Day 14
Control	181.00 ± 4.00	184.00 ± 5.34	192.00 ± 3.74
*C. papaya* L. Leaf Extract 5000 mg/kg	180.00 ± 5.70	192.00 ± 3.00	202.00 ± 4.90

Values are expressed as mean ± S.E.M., *n* = 5 (female).

**Table 6 toxics-12-00198-t006:** Organ weight (g) of female rats in the acute toxicity test.

Organs	Female	
	Control	*Carica papaya* L. Extract (5000 mg/kg)
Brain	2.03 ± 0.08	2.08 ± 0.07
Lung	0.96 ± 0.07	1.10 ± 0.13
Heart	0.63 ± 0.06	0.73 ± 0.12
Liver	7.90 ± 0.83	8.60 ± 1.22
Spleen	0.45 ± 0.07	0.52 ± 0.04
Adrenal gland	0.03 ± 0.01	0.04 ± 0.01
Kidney	0.97 ± 0.05	1.13 ± 0.14
Ovary	0.08 ± 0.03	0.07 ± 0.02
Uterus	0.89 ± 0.51	0.47 ± 0.04 *

Values are expressed as mean ± S.E.M.; *n* = 5 (female). * Statistically significant difference compared to the control group (*p* < 0.05).

**Table 7 toxics-12-00198-t007:** Longitudinal body weight profiles of female rats subjected to a six-month chronic toxicity test with *C. papaya* leaf extract.

Day	Control	Satellite Control	*Carica papaya* L. Extract (mg/kg)				
			100	400	1000	5000	Satellite
1	180.50 ± 6.60	171.00 ± 7.81	184.50 ± 6.89	189.50 ± 5.55	187.00 ± 5.07	188.00 ± 5.18	156.00 ± 5.10
30	242.50 ± 5.64	237.00 ± 12.00	238.50 ± 5.17	247.50 ± 5.12	250.00 ± 3.07	248.50 ± 2.59	247.00 ± 6.44
45	264.00 ± 4.27	265.00 ± 8.94	255.00 ± 3.65	265.50 ± 5.03	274.00 ± 4.52	266.00 ± 5.10	264.00 ± 1.00
60	263.50 ± 4.22	270.00 ± 11.40	261.50 ± 3.80	275.50 ± 4.62	274.00 ± 4.40	272.00 ± 4.73	273.00 ± 5.39
90	283.50 ± 5.11	288.00 ± 9.57	280.00 ± 4.53 *	292.00 ± 4.30	291.00 ± 4.27	285.00 ± 3.87	297.00 ± 6.44
120	292.50 ± 4.10	301.00 ± 8.86	281.50 ± 3.42	299.00 ± 5.67	299.00 ± 4.27	292.50 ± 4.79	308.00 ± 11.25
150	289.50 ± 5.94	290.00 ± 9.35	284.00 ± 3.23	292.50 ± 5.12	294.50 ± 5.08	292.50 ± 6.84	302.00 ± 11.58
180	287.50 ± 5.44	284.00 ± 11.55	281.50 ± 2.69	292.50 ± 5.59	295.50 ± 5.65	295.00 ± 7.34	304.00 ± 14.27

Values are mean ± S.E.M., *n* = 10, *n* = 5 (satellite groups). * Statistically significant difference compared to the control group (*p* < 0.05).

**Table 8 toxics-12-00198-t008:** Longitudinal body weight profiles of male rats subjected to a six-month chronic toxicity test with *C. papaya* leaf extract.

Day	Control	Satellite Control	*Carica papaya* L. Extract (mg/kg)				
			100	400	1000	5000	Satellite
1	180.50 ± 6.60	184.00 ± 4.85	184.50 ± 4.97	184.50 ± 4.86	182.50 ± 3.59	187.00 ± 3.74	183.00 ± 4.34
30	352.00 ± 8.54	336.00 ± 15.28	354.50 ± 6.64	356.00 ± 4.76	359.50 ± 7.32	362.50 ± 5.59	321.00 ± 18.13 *
45	407.50 ± 8.17	416.00 ± 7.31	401.00 ± 8.56	399.50 ± 8.93	396.50 ± 11.90	408.00 ± 8.47	448.00 ± 20.65
60	419.00 ± 8.88	427.00 ± 3.74	413.00 ± 10.39	411.50 ± 10.03	414.00 ± 12.29	422.00 ± 8.57	472.00 ± 23.91
90	447.00 ± 10.28	460.00 ± 10.49	442.00 ± 10.47	441.00 ± 11.16	440.00 ± 15.72	443.00 ± 15.04	490.00 ± 34.53
120	458.00 ± 13.38	460.00 ± 11.07	454.50 ± 12.50	457.50 ± 8.51	451.50 ± 13.83	466.00 ± 16.63	506.00 ± 26.52
150	453.50 ± 15.44	463.00 ± 8.75	449.50 ± 11.77	453.50 ± 8.63	434.50 ± 15.77	474.00 ± 18.77	525.40 ± 43.71
180	468.00 ± 14.22	486.00 ± 13.27	466.50 ± 15.22	475.00 ± 8.66	468.00 ± 23.13	488.00 ± 22.86	545.00 ± 40.99

Values are mean ± S.E.M., *n* = 10, *n* = 5 (satellite groups). * Statistically significant difference compared to the control group (*p* < 0.05).

**Table 9 toxics-12-00198-t009:** Organ weights of female rats subjected to a six-month chronic toxicity test with *C. papaya* leaf extract.

Organ	Control	Satellite Control	*Carica papaya* L. Extract (mg/kg)				
			100	400	1000	5000	Satellite 5000 mg/kg bw
Brain	2.18 ± 0.03	2.31 ± 0.05	2.21 ± 0.04	2.22 ± 0.05	2.15 ± 0.03	2.27 ± 0.04	2.14 ± 0.07
Lung	1.69 ± 0.06	2.55 ± 0.42	1.91 ± 0.14	1.97 ± 0.16	1.75 ± 0.08	1.76 ± 0.06	1.90 ± 0.23
Heart	0.99 ± 0.04	1.02 ± 0.04	0.97 ± 0.03	0.99 ± 0.02	0.95 ± 0.02	1.00 ± 0.03	1.08 ± 0.03
Liver	11.18 ± 0.25	11.90 ± 0.43	10.94 ± 0.34	11.48 ± 0.38	11.12 ± 0.36	11.82 ± 0.46	12.43 ± 0.75
Spleen	0.58 ± 0.09	0.46 ± 0.11	0.50 ± 0.02	0.45 ± 0.05	0.49 ± 0.02	0.53 ± 0.03	0.54 ± 0.04
Adrenal gland	0.04 ± 0.00	0.04 ± 0.00	0.04 ± 0.00	0.04 ± 0.00	0.04 ± 0.00	0.04 ± 0.00	0.03 ± 0.00
Kidney	1.32 ± 0.04	1.29 ± 0.09	1.28 ± 0.05	1.27 ± 0.04	1.27 ± 0.04	1.38 ± 0.04	1.45 ± 0.03
Ovary	0.06 ± 0.01	0.09 ± 0.04	0.07 ± 0.01	0.07 ± 0.01	0.06 ± 0.01	0.07 ± 0.01	0.07 ± 0.00
Uterus	1.31 ± 0.15	1.38 ± 0.21	1.31 ± 0.18	1.19 ± 0.13	1.03 ± 0.07	1.48 ± 0.23	1.58 ± 0.16

Values are mean ± S.E.M., *n* = 10, *n* = 5 (satellite groups).

**Table 10 toxics-12-00198-t010:** Organ weights of male rats subjected to a six-month chronic toxicity test with *C. papaya* leaf extract.

Organs	Control	Satellite Control	*Carica papaya* L. Extract (mg/kg)				
			100	400	1000	5000	Satellite 5000 mg/kg bw
Brain	2.39 ± 0.03	2.41 ± 0.03	2.44 ± 0.03	2.40 ± 0.04	2.38 ± 0.04	2.41 ± 0.04	2.41 ± 0.07
Lung	2.80 ± 0.39	2.43 ± 0.33	2.19 ± 0.10	2.31 ± 0.08	2.32 ± 0.22	2.41 ± 0.10	1.94 ± 0.35
Heart	1.25 ± 0.04	1.17 ± 0.04	1.21 ± 0.06	1.27 ± 0.03	1.25 ± 0.05	1.30 ± 0.06	1.38 ± 0.10
Liver	13.98 ± 0.63	13.40 ± 0.50	12.49 ± 1.52	14.78 ± 0.46	13.36 ± 0.83	14.44 ± 0.88	15.72 ± 1.01
Spleen	0.64 ± 0.04	0.72 ± 0.01	0.68 ± 0.04	0.69 ± 0.02	0.68 ± 0.05	0.68 ± 0.04	0.82 ± 0.04
Adrenal gland	0.03 ± 0.00	0.03 ± 0.00	0.03 ± 0.00	0.07 ± 0.04	0.03 ± 0.00	0.03 ± 0.00	0.03 ± 0.00
Kidney	1.58 ± 0.06	1.65 ± 0.05	1.63 ± 0.08	1.67 ± 0.05	1.62 ± 0.07	1.81 ± 0.08	1.97 ± 0.21
Epididymis	0.72 ± 0.03	0.80 ± 0.04	0.72 ± 0.02	0.76 ± 0.03	0.77 ± 0.04	0.81 ± 0.01	0.86 ± 0.04
Testis	1.61 ± 0.13	1.70 ± 0.04	1.64 ± 0.11	1.75 ± 0.12	1.70 ± 0.14	1.78 ± 0.07	1.90 ± 0.08

Values are mean ± S.E.M., *n* = 10, *n* = 5 (satellite groups).

**Table 11 toxics-12-00198-t011:** Hematological profile of female rats subjected to six-month chronic toxicity test with *C. papaya* leaf extract.

Blood Parameters	Control	Satellite Control	*Carica papaya* L. Extract (mg/kg)				Satellite 5000 mg/kg bw
			100	400	1000	5000	
Red blood cell (×10^6^/μL)	8.05 ± 0.12	7.03 ± 0.40	7.94 ± 0.15	7.85 ± 0.07	7.98 ± 0.12	7.63 ± 0.13	7.35 ± 0.32
Hemoglobin (g/dL)	14.87 ± 0.19	13.08 ± 0.65	14.68 ± 0.24	14.77 ± 0.14	15.10 ± 0.18	14.24 ± 0.24	13.86 ± 0.42
Hematocrit (%)	43.53 ± 0.54	40.02 ± 0.90	43.38 ± 0.77	43.05 ± 0.29	44.13 ± 0.66	41.70 ± 0.71	40.98 ± 1.15
Mean corpuscular volume (fL)	54.15 ± 0.38	54.40 ± 0.39	54.63 ± 0.44	54.38 ± 0.34	56.45 ± 1.34	54.69 ± 0.37	55.92 ± 1.32
Mean corpuscular hemoglobin (pg)	18.48 ± 0.17	18.64 ± 0.21	18.51 ± 0.17	18.67 ± 0.16	18.95 ± 0.18	18.67 ± 0.14	18.92 ± 0.39
Mean corpuscular hemoglobin concentration (g/dL)	34.20 ± 0.14	34.28 ± 0.23	33.87 ± 0.14	34.30 ± 0.15	34.24 ± 0.15	34.16 ± 0.16	33.84 ± 0.13
Platelet (×10^5^/μL)	9.66 ± 0.31	7.89 ± 0.33	10.05 ± 1.11	9.13 ± 0.46	10.40 ± 0.75	10.15 ± 0.60	7.20 ± 1.06
White blood cells (×10^3^/μL)	3.03 ± 0.25	3.23 ± 0.68	3.35 ± 0.45	3.82 ± 0.23	3.69 ± 0.41	4.04 ± 0.70	3.09 ± 0.29
Neutrophil (×10^3^/μL)	0.92 ± 0.08	1.19 ± 0.25	0.99 ± 0.20	1.18 ± 0.10	1.02 ± 0.14	1.01 ± 0.18	1.38 ± 0.42
Lymphocyte (×10^3^/μL)	1.63 ± 0.19	1.76 ± 0.37	1.98 ± 0.30	2.28 ± 0.15	2.24 ± 0.09	2.43 ± 0.18	1.53 ± 0.29
Monocyte (×10^3^/μL)	0.18 ± 0.04	0.20 ± 0.05	0.28 ± 0.07	0.25 ± 0.03	0.35 ± 0.07	0.30 ± 0.11	0.20 ± 0.05
Eosinophil (×10^3^/μL)	0.10 ± 0.02	0.07 ± 0.02	0.10 ± 0.02	0.12 ± 0.01	0.08 ± 0.02	0.11 ± 0.03	0.05 ± 0.01
Basophil (×10^3^/μL)	0.00 ± 0.00	0.00 ± 0.00	0.00 ± 0.00	0.00 ± 0.00	0.00 ± 0.00	0.00 ± 0.00	0.00 ± 0.00

Values are mean ± S.E.M., *n* = 10, *n* = 5 (satellite groups).

**Table 12 toxics-12-00198-t012:** Hematology profile of male rats treated with *C. papaya* leaf extract in the six-month chronic toxicity test.

Blood Parameters	Control	Satellite Control	*Carica papaya* L. Extract (mg/kg)				Satellite 5000 mg/kg bw
			100	400	1000	5000	
Red blood cell (×10^6^/μL)	8.79 ± 0.18	9.53 ± 0.82	8.76 ± 0.15	8.74 ± 0.14	8.57 ± 0.14	8.74 ± 0.18	8.47 ± 0.44
Hemoglobin (g/dL)	15.52 ± 0.29	16.64 ± 1.13	15.44 ± 0.15	15.40 ± 0.21	15.28 ± 0.13	15.23 ± 0.20	14.92 ± 0.68
Hematocrit (%)	46.27 ± 0.90	49.88 ± 4.37	46.22 ± 0.58	45.77 ± 0.67	45.42 ± 0.38	45.23 ± 0.80	44.16 ± 2.06
Mean corpuscular volume (fL)	52.63 ± 0.40	52.34 ± 0.76	52.84 ± 0.60	52.42 ± 0.80	52.68 ± 0.58	51.79 ± 0.48	52.16 ± 0.44
Mean corpuscular hemoglobin (pg)	17.66 ± 0.11	17.70 ± 0.23	17.65 ± 0.22	17.64 ± 0.26	17.71 ± 0.18	17.48 ± 0.23	17.64 ± 0.18
Mean corpuscular hemoglobin concentration (g/dL)	33.53 ± 0.14	33.70 ± 0.30	33.42 ± 0.17	33.64 ± 0.23	33.64 ± 0.09	33.70 ± 0.24	33.80 ± 0.24
Platelet (×10^5^/μL)	8.27 ± 0.29	7.84 ± 0.36	8.33 ± 0.15	8.55 ± 0.29	8.20 ± 0.31	8.87 ± 0.28	7.82 ± 0.76
White blood cells (×10^3^/μL)	5.05 ± 0.54	4.66 ± 0.52	5.38 ± 0.48	5.06 ± 0.36	5.32 ± 0.61	5.09 ± 0.55	4.68 ± 0.96
Neutrophil (×10^3^/μL)	1.36 ± 0.17	0.74 ± 0.12	1.49 ± 0.17	1.44 ± 0.16	1.39 ± 0.23	1.57 ± 0.21	1.08 ± 0.31
Lymphocyte (×10^3^/μL)	3.26 ± 0.37	2.27 ± 0.42	3.52 ± 0.32	3.41 ± 0.21	3.44 ± 0.36	3.07 ± 0.33	3.10 ± 0.66
Monocyte (×10^3^/μL)	0.31 ± 0.05	0.35 ± 0.09	0.25 ± 0.01	0.27 ± 0.03	0.35 ± 0.07	0.29 ± 0.06	0.39 ± 0.07
Eosinophil (×10^3^/μL)	0.13 ± 0.02	0.22 ± 0.12	0.13 ± 0.01	0.16 ± 0.02	0.15 ± 0.03	0.17 ± 0.03	0.12 ± 0.02
Basophil (×10^3^/μL)	0.00 ± 0.00	0.00 ± 0.00	0.00 ± 0.00	0.00 ± 0.00	0.00 ± 0.00	0.00 ± 0.00	0.00 ± 0.00

Values are mean ± S.E.M., *n* = 10, *n* = 5 (satellite groups).

**Table 13 toxics-12-00198-t013:** Biochemical parameters of female rats in the chronic toxicity test.

Blood Parameters	Control	Satellite Control	*Carica papaya* L. Extract (mg/kg)				Satellite 5000 mg/kg bw
			100	400	1000	5000	
BUN (mg/dL)	18.22 ± 1.12	17.36 ± 1.43	16.11 ± 0.85	16.85 ± 0.70	17.81 ± 1.15	18.51 ± 1.23	19.78 ± 2.60
Creatinine (mg/dL)	0.75 ± 0.02	0.74 ± 0.02	0.74 ± 0.02	0.74 ± 0.03	0.75 ± 0.04	0.75 ± 0.03	0.74 ± 0.02
Total protein (g/dL)	7.24 ± 0.10	7.38 ± 0.34	7.27 ± 0.14	7.15 ± 0.07	7.08 ± 0.18	7.23 ± 0.12	7.46 ± 0.24
Albumin (g/dL)	3.73 ± 0.06	3.86 ± 0.17	3.62 ± 0.10	3.67 ± 0.05	3.62 ± 0.11	3.68 ± 0.06	3.72 ± 0.24
Total bilirubin (mg/dL)	0.17 ± 0.01	0.13 ± 0.02	0.15 ± 0.01	0.15 ± 0.00	0.16 ± 0.01	0.14 ± 0.01	0.13 ± 0.01
Direct bilirubin (mg/dL)	0.08 ± 0.01	0.07 ± 0.01	0.08 ± 0.01	0.07 ± 0.00	0.07 ± 0.00	0.07 ± 0.00	0.08 ± 0.00
AST (U/L)	101.10 ± 7.17	124.80 ± 25.06	90.30 ± 5.87	97.00 ± 5.09	90.90 ± 6.38	89.40 ± 7.18	120.40 ± 13.50
ALT (U/L)	24.80 ± 2.89	29.40 ± 2.87	27.00 ± 1.97	25.50 ± 1.88	27.10 ± 2.27	27.00 ± 2.61	29.80 ± 3.44
ALP (U/L)	23.40 ± 1.39	25.20 ± 2.63	28.50 ± 4.63	26.20 ± 2.60	25.40 ± 1.49	25.50 ± 1.86	26.60 ± 3.58
Glucose (mg/dL)	126.58 ± 2.59	155.20 ± 11.66	149.08 ± 9.54	127.50 ± 2.66	124.42 ± 3.23	130.27 ± 7.90	142.80 ± 12.48

BUN, blood urea nitrogen; AST, aspartate aminotransferase; ALT, alanine aminotransferase, ALP alkaline phosphatase. Values are mean ± S.E.M., *n* = 10, *n* = 5 (satellite groups).

**Table 14 toxics-12-00198-t014:** Biochemical parameters of male rats in the chronic toxicity test.

Blood Parameters	Control	Satellite Control	*Carica papaya* L. Extract (mg/kg)				Satellite 5000 mg/kg bw
			100	400	1000	5000	
BUN (mg/dL)	18.13 ± 0.81	16.06 ± 0.67	17.53 ± 0.84	19.10 ± 1.78	18.53 ± 0.97	16.91 ± 1.19	14.20 ± 0.60
Creatinine (mg/dL)	0.60 ± 0.02	0.60 ± 0.03	0.62 ± 0.02	0.59 ± 0.01	0.59 ± 0.01	0.60 ± 0.01	0.56 ± 0.03
Total protein (g/dL)	6.06 ± 0.13	5.92 ± 0.11	6.13 ± 0.09	6.19 ± 0.11	6.02 ± 0.05	6.29 ± 0.10	5.82 ± 0.11
Albumin (g/dL)	2.80 ± 0.04	2.76 ± 0.02	2.81 ± 0.06	2.94 ± 0.04	2.84 ± 0.04	2.86 ± 0.06	2.84 ± 0.07
Total bilirubin (mg/dL)	0.11 ± 0.01	0.11 ± 0.02	0.12 ± 0.01	0.11 ± 0.01	0.11 ± 0.01	0.11 ± 0.01	0.10 ± 0.01
Direct bilirubin mg/dL)	0.07 ± 0.00	0.07 ± 0.01	0.07 ± 0.010	0.06 ± 0.00	0.07 ± 0.01	0.06 ± 0.00	0.07 ± 0.01
AST (U/L)	129.60 ± 16.26	112.00 ± 18.84	135.00 ± 10.90	143.00 ± 8.67	118.60 ± 7.92	133.40 ± 6.49	135.40 ± 29.38
ALT (U/L)	39.70 ± 9.57	31.80 ± 5.86	35.20 ± 6.08	35.40 ± 3.39	35.80 ± 4.59	33.80 ± 1.79	30.00 ± 6.54
ALP (U/L)	88.10 ± 6.07	63.80 ± 4.22	81.20 ± 3.14	80.60 ± 5.72	75.30 ± 5.74	74.80 ± 6.08	52.80 ± 3.92 *
Glucose (mg/dL)	130.08 ± 7.35	122.33 ± 6.49	133.17 ± 6.63	144.50 ± 6.43	133.83 ± 6.79	126.91 ± 13.72	149.60 ± 15.88

BUN, blood urea nitrogen; AST, aspartate aminotransferase; ALT, alanine aminotransferase, ALP alkaline phosphatase. Values are mean ± S.E.M., *n* = 10, *n* = 5 (satellite groups). * Statistically significant difference compared to the control group (*p* < 0.05).

## Data Availability

Data is available upon request.

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
