# Peer review of "Safety of Oral Carica papaya L. Leaf 10% Ethanolic Extract for Acute and Chronic Toxicity Tests in Sprague Dawley Rats"

_toxics, 2024, doi:10.3390/toxics12030198_

Round 1

Reviewer 1 Report (New Reviewer)

Comments and Suggestions for Authors

Line 100: how did the authors clean the leaves?

Line 102: the leaves were drying for how long?

Line 103-105: the authors should provide more details here. What was the model of the sieve grinder? What was the particle size? What was the "cool, dry environment"?

Line 106-107: 10% is v/v? Is it a water-ethanol solution? What was the condition of "agitated continuously "? Please provide the details of filtration.

Line 109: how to adjust the weight of the solution?

Line 110-113: what was the ratio between maltodextrin and the extract? How long did they mix? What was the model of the spray dryer? How was the flow speed?

Line 186: why did the authors not use only SD or SEM? Why did not use the same data analysis method?

Figure 1: the authors should not use the same Y axile for both samples in one figure, which will change the value of response values. Different samples should be in their own figure.

Table 6: the authors should add the full illustration of the data analysis in the caption. The same scenario is in all figures with significance analysis throughout the text.

Author Response

We are very grateful to the editors and reviewers for your serious and responsible review to our manuscript and thank you very much for giving us valuable comments and suggestions, which have important guiding significance for improving our writing level and future scientific research. Now I will answer one by one based on the editor's and reviewers' comments.

Comments and Suggestions for Authors

Line 100: how did the authors clean the leaves?

Response: Thank you very much for your remark. The leaves of C. papaya were initially cleaned with cleaning cloths to wipe them, depurated of their central stems and large branches, followed by triple rinsing with distilled water.

Line 102: the leaves were drying for how long?

Response: Thank you very much for your remark. The cleaned leaves were then subjected to hot air oven drying at 50 °C for 12 hours.

Line 103-105: the authors should provide more details here. What was the model of the sieve grinder? What was the particle size? What was the "cool, dry environment"?

Response: Thank you very much for your remark. The dried material was pulverized using an 80-mesh sieve grinder (Nanotech NT-1000D; Thanapan Ltd., Chiang Mai, Thailand) to achieve a homogeneous particle size of less than 177 microns, maximizing the surface area for optimal extraction efficiency. This processed leaf powder was securely stored in a vacuum-sealed bag protected from light and stored at -20°C.

Line 106-107: 10% is v/v? Is it a water-ethanol solution? What was the condition of "agitated continuously "? Please provide the details of filtration.

Response: Thank you very much for your remark. For the extraction process, 8 kg of C. papaya leaf powder was prepared in a 60 L solution of 10% v/v ethanol in water, using high-speed extraction (EURO Best Technology Co., Ltd., Bangkok, Thailand) at 150 rpm and a temperature of 80–90 °C for 3 hours.

Line 109: how to adjust the weight of the solution?

Response: The separated solution was then weighed and adjusted to a total of 150 kg using 10%v/v ethanol in water solution.

Line 110-113: what was the ratio between maltodextrin and the extract? How long did they mix? What was the model of the spray dryer? How was the flow speed?

Response: Thank you very much for your remark. The concentration of maltodextrin was 0.5%w/v and homogenized at 5,000 rpm. Then, the obtained solution was spray dry (SDE-50 EURO Best Technology Co., Ltd., Bangkok, Thailand) with an inlet temperature of 160 oC and the flow rate was set at 500 mL/min.

Line 186: why did the authors not use only SD or SEM? Why did not use the same data analysis method?

Response: We apologize for the error in Table 6; it should be SEM, not SD. However, for consistency in all values presented in the manuscript, the author has opted for the mean with SEM. This choice is made as SEM measures the precision of the population estimate.

Figure 1: the authors should not use the same Y axile for both samples in one figure, which will change the value of response values. Different samples should be in their own figure.

Response: Thank you for your pointing out. We have modified the figure 1 by separating between standards and sample.

Table 6: the authors should add the full illustration of the data analysis in the caption. The same scenario is in all figures with significance analysis throughout the text.

Response: Thank you for your pointing out. We have added the illustration and the meaning of its symbols as other figures or tables.

Reviewer 2 Report (New Reviewer)

Comments and Suggestions for Authors

The manuscript, 'Safety of Oral Carica papaya L. Leaf 10% Ethanolic Extract for 2 Acute and Chronic Toxicity Tests in Sprague-Dawley Rats,' is appropriately written. It thoroughly explains the research undertaken. The examinations were well-planned and documented in detail. The results are stated briefly and concisely. However, I occasionally overlook details, such as what was examined in histopathology preparations. After conducting a more thorough study, I see no changes, so I understand the abrupt exclusion of a complete description, which would be valuable, for example, in supplemental data.

Was the amount of water and food intake assessed? Was the amount and color of urine assessed?

‘Caricaseae’ (41 line), ‘in vivo’ (71 line), ad libitum (134 line) should be in italics

Line 75: ‘mg’ or ‘mg/kg’?

Author Response

We are very grateful to the editors and reviewers for your serious and responsible review to our manuscript and thank you very much for giving us valuable comments and suggestions, which have important guiding significance for improving our writing level and future scientific research. Now I will answer one by one based on the editor's and reviewers' comments.

Comments and Suggestions for Authors

The manuscript, 'Safety of Oral Carica papaya L. Leaf 10% Ethanolic Extract for 2 Acute and Chronic Toxicity Tests in Sprague-Dawley Rats,' is appropriately written. It thoroughly explains the research undertaken. The examinations were well-planned and documented in detail. The results are stated briefly and concisely. However, I occasionally overlook details, such as what was examined in histopathology preparations. After conducting a more thorough study, I see no changes, so I understand the abrupt exclusion of a complete description, which would be valuable, for example, in supplemental data.

Response: We appreciate your suggestion. According to the WHO guideline, specifically the "General Guidelines for Methodologies on Research and Evaluation of Traditional Medicine" in Annex II on page 31, and OECD 452 on page 15, it is imperative to conduct histopathological examinations of vital organs and present the findings in the report. This practice is essential to ensure the safety of the test substance.

Was the amount of water and food intake assessed? Was the amount and color of urine assessed?

Response: In WHO guideline (General Guidelines for Methodologies on Research and Evaluation of Traditional Medicine) on page 30, states that "For all experimental animals, the general signs should be observed daily, and body weight and food intake should be measured periodically. If useful, water intake should also be determined." and OECD guideline 452, pages 14, it is noted that water and food consumption can be reported as if applicable. And throughout the experiment, the animals' health, as well as their water and food consumption, were monitored on every day, and all results were found to be normal. Thus, the results of food and water consumption were not reported in our manuscript. In conducting urine examination, the researchers focused on general toxicity rather than toxicokinetics; consequently, urine collection was not undertaken in this study. Nevertheless, it is important to note that general observations of urination in the animals remained normal throughout the study.

‘Caricaseae’ (41 line), ‘in vivo’ (71 line), ad libitum (134 line) should be in italics

Response: Thank you for your remark. We have carefully checked and edited those words and others in our manuscript.

Line 75: ‘mg’ or ‘mg/kg’?

Response: We sincerely apologize for the error. We have corrected this word.

Reviewer 3 Report (New Reviewer)

Comments and Suggestions for Authors

In this study, the authors repeat the effect of extracted C. papaya leaf in Sprague-Dawley rats. Generally, papaya itself is a commonly used safe food for humans. Therefore, this study is nothing new as explained below.

1. As the citation in Introduction [2-5], C. papaya leaf extract is rich in bioactive components such as flavonoids, alkaloids, saponins, phytosterols, phenolics, and tannins, highlighting its prospective pharmaceutical utility. The components of the extracts in this study differed from those in the previous study (Afzan et al., 2012). In addition, the extraction process will lose the real active components, not just the drying loss in hot air oven drying at 50℃. The loss might be the critical effective factor in cytotoxicity.

2. Depending on the experimental procedure, the authors gave rats a range of extraction doses up to 5,000 mg/kg body weight (= 5 g/kg). However, the maximum dose of 5 g/kg is only twice the dose cited in previous studies (Afzan et al., 2012), so a single dose of 5 g/kg not producing significant toxicity should be expected.

3. Since papaya is a common natural food, a daily dose of 5 g/kg for 180 days should be similar to previous studies that lasted only 28 days. The authors provided more additional results in conspecific rats, but the results showing no toxicity should still be consistent with expectations. Therefore, this study is not supposed to be a new report.

Afzan, A., Abdullah, N.R., Halim, S.Z., Rashid, B.A., Semail, R.H., Abdullah, N., Jantan, I., Muhammad, H., Ismail, Z., 2012. Repeated dose 28-days oral toxicity study of Carica papaya L. leaf extract in Sprague Dawley rats. Molecules 17, 4326-4342.

Comments on the Quality of English Language

Minor editing of English language required

Author Response

We are very grateful to the editors and reviewers for your serious and responsible review to our manuscript and thank you very much for giving us valuable comments and suggestions, which have important guiding significance for improving our writing level and future scientific research. Now I will answer one by one based on the editor's and reviewers' comments.

Comments and Suggestions for Authors

In this study, the authors repeat the effect of extracted C. papaya leaf in Sprague-Dawley rats. Generally, papaya itself is a commonly used safe food for humans. Therefore, this study is nothing new as explained below.

  1. As the citation in Introduction [2-5], C. papaya leaf extract is rich in bioactive components such as flavonoids, alkaloids, saponins, phytosterols, phenolics, and tannins, highlighting its prospective pharmaceutical utility. The components of the extracts in this study differed from those in the previous study (Afzan et al., 2012). In addition, the extraction process will lose the real active components, not just the drying loss in hot air oven drying at 50℃. The loss might be the critical effective factor in cytotoxicity.

Response: The study suggests that plants, both within the same region and across different regions, exhibit significant quantitative and qualitative impacts on their chemical profiles [1]. These variations seem to be associated with different extraction methods, leading to a diverse range of substance compositions. Furthermore, the use of various mobile phases in HPLC analysis contributes to different separation outcomes. Therefore, standardizing raw materials and extracts is crucial.

In this study, we have adopted the quality control methods outlined in the Thai Herbal Pharmacopoeia. The quality of the extracts in this study has been meticulously monitored, considering both the raw materials and the final extracts. The pharmacological activities of this extract, specifically its effects on platelet aggregation, have been examined. To prepare the data for submission to Human Ethical Committees for the approval of human testing in the clinical phase, it is imperative that the general toxicology tests utilize the extract with the same quality control measures and characteristics as those employed in the pharmacological studies. Crucially, this experiment aims to verify and ensure the safety of an extract that maintains consistent properties when tested for pharmacological effects.

Reference: 1. de Sá Filho, José Carlos Freitas, et al. "Geographic location and seasonality affect the chemical composition of essential oils of Lippia alba accessions." Industrial Crops and Products 188 (2022): 115602.

  1. Depending on the experimental procedure, the authors gave rats a range of extraction doses up to 5,000 mg/kg body weight (= 5 g/kg). However, the maximum dose of 5 g/kg is only twice the dose cited in previous studies (Afzan et al., 2012), so a single dose of 5 g/kg not producing significant toxicity should be expected.

Response: Thank you for your comment. The selection of the 5,000 mg/kg dose for the acute toxicity test was based on the guidelines of the OECD 420 Acute Toxicity Tests (Annex. 4, pages 13-14). These guidelines recommend administering a dose of either 2,000 mg/kg or 5,000 mg/kg to the initial test animal when the substance is presumed relatively safe, followed by a 48-hour observation period for potential toxicity.

The purpose of this testing is to establish the maximum tolerable dose of the extract, a critical step in ensuring its safety before proceeding to clinical trials. The decision to use a dose of 5,000 mg/kg was in alignment with these OECD recommendations. Importantly, the efficacy of this extract had already been established in a preceding pharmacological study. The selection of the 5,000 mg/kg dose not only adheres to the recommended guidelines but also aids in determining the therapeutic index of the extract, providing a comprehensive safety profile essential for future human studies.

  1. Since papaya is a common natural food, a daily dose of 5 g/kg for 180 days should be similar to previous studies that lasted only 28 days. The authors provided more additional results in conspecific rats, but the results showing no toxicity should still be consistent with expectations. Therefore, this study is not supposed to be a new report.

Response: Thank you for your pointing out. The focus of our research is on the use of Carica papaya within Thailand, where the fruit is commonly consumed but the leaves are not traditionally used as food or dietary supplements. This contrasts with the study by Afzan et al., which examines papaya leaves in Malaysia, employing a distinct extraction method. Specifically, Afzan et al. used a process involving juice extraction followed by lyophilization. In contrast, our study adopts a novel approach by employing fermentation with 10% ethanol for 3 hours, followed by spray drying.

In alignment with the WHO guidelines (General Guidelines for Methodologies on Research and Evaluation of Traditional Medicine, 2000) as noted on page 29, the duration of toxicity testing in animals is directly correlated with the expected period of clinical use or consumption of the substance. The guideline recommends that if the substance is expected to be consumed for a period ranging from 1 to 6 months, animal testing should extend for a duration of 3 to 6 months. The toxicity testing in Afzan's study was conducted for a duration of 28 days, presuming that the expected period of clinical use or consumption of the substance would be less than one week.

Our study adheres to WHO guideline, ensuring that the duration of our toxicity testing aligns with the expected period of human consumption, thereby providing a robust and reliable safety profile. In consideration of these points, it is evident that our study provides distinct and valuable insights, differentiating it from the study conducted by Afzan and colleagues. Thus, our research contributes novel findings to the scientific understanding of Carica papaya's effects and its potential applications.

Afzan, A., Abdullah, N.R., Halim, S.Z., Rashid, B.A., Semail, R.H., Abdullah, N., Jantan, I., Muhammad, H., Ismail, Z., 2012. Repeated dose 28-days oral toxicity study of Carica papaya L. leaf extract in Sprague Dawley rats. Molecules 17, 4326-4342.

Comments on the Quality of English Language

Minor editing of English language required

Response: Thank you for your pointing out. We carefully reviewed the wording and grammatical errors.

Round 2

Reviewer 3 Report (New Reviewer)

Comments and Suggestions for Authors

As the author responds to comment #1, this is just a report that needs to be submitted to the Human Ethics Committee for approval of human testing. In this report, the authors extracted Carica papaya L. Leaf using 10% ethanol and tested its safety in rats. While safety is expected, the authors emphasize that the extraction process is superior to that used in other studies. However, the comparison between this and other methods is not discussed in detail here. In addition, this method may eliminate the potential active ingredient in the extract, so there are no significant negative effects, as shown in this report. Finally, this manuscript needs to be checked, as the author still has some errors, such as "S.E.M" or "S.M.E" with limited revisions. Therefore, this should not be enough for this journal.

Comments on the Quality of English Language

The author only revised a limited content in “Materials and Methods” and Tables.

This manuscript is a resubmission of an earlier submission. The following is a list of the peer review reports and author responses from that submission.

Round 1

Reviewer 1 Report

Comments and Suggestions for Authors

The manuscript is interesting, but my suggestion is that the authors should explain why they chose an ethanolic extract?

 In the Materials and methods section, describe how the rats were sacrificed.

In the Results section, the data from Table 1 must be included in the text and not in the table. Similarly, Table 6

The interpretations of the Hippocratic Assessment (Decrease of motor activity,  Decrease of respiration rate, Loss of righting reflex, Loss of screen grip)  should be described in the Material and methods section

Author Response

Response to Reviewer 1

We are very grateful to the editors and reviewers for your serious and responsible review to our manuscript and thank you very much for giving us valuable comments and suggestions, which have important guiding significance for improving our writing level and future scientific research. Now I will answer one by one based on the editor's and reviewers' comments.   

 Comments and Suggestions for Authors

The manuscript is interesting, but my suggestion is that the authors should explain why they chose an ethanolic extract?

Response: We have explained more in this response. Our group discovered in a prior investigation that a 10% ethanol extract exhibited favorable pharmacological effects (unpublished data) in relation to the subject aspect of "Implications for Patients with Thrombocytopenia: Research and Development of Papaya Extract." This research can provide scientific support for the Thai FDA registration of papaya extract safety data. As a result, 10% ethanolic extracts have been used in this investigation in accordance with above-mentioned pharmacological activity tests. The registration requirements of the Thai FDA also state that the efficacy and toxicity tests must use the same extract. But this information will not be included in our manuscript.

 In the Materials and methods section, describe how the rats were sacrificed.

Response: Thank you very much for pointing this out. The process for sacrificing rats has been added to the Materials and Methods section of topic 2.7. All rats were euthanized using intraperitoneal thiopental sodium injection at a dose of 120 mg/kg. Moreover, after euthanasia, animals need to undergo a physical examination by checking their vital signs, pulse, and reflexes to confirm death.

In the Results section, the data from Table 1 must be included in the text and not in the table. Similarly, Table 6.

 Response: Your pointing out is greatly appreciated. The results section, which is associated with Table 1, has been modified to include the appropriate information in the text.

The interpretations of the Hippocratic Assessment (Decrease of motor activity, Decrease of respiration rate, Loss of righting reflex, Loss of screen grip) should be described in the Material and methods section.

Response: Thank you very much for pointing this out. The Hippocratic assessment has been modified in topic 2.6 for acceptable data.

Reviewer 2 Report

Comments and Suggestions for Authors

In this manuscript authors described the safety profile of C. papaya leaf ethanolic extract  in Sprague-Dawley rats. The authors performed the chemical analysis of extract and acute and chronic toxicity studies with different dosses of the extract in rats. The acute toxicity test was performed by using a single oral dose of 5000 mg/kg body weight and the chronic toxicity study by using daily oral doses of 100, 400, 1000, and 5000 mg/kg over 180 days. Final results of the tests have shown that C. papaya leaf extract exhibited no adverse effects on the rats during these studies.

This study is important because it gives a new data on the safety profile of C. papaya leaf ethanolic extract with higher doses.

The tests are well designed and executed and the obtained data is presented in a clear and coherent manner.

The methods applied are adequately described.

The final concluding remarks are appropriate and supported by the data presented in the manuscript.

The references cited are relevant and sufficient.

Tables and figures are well prepared. 

The manuscript is well written and easy to read even though the manuscript needs some additions/corrections (described below):

Line 2 – in the title it should also stay Leaf  … Carica papaya L. Leaf ethanolic extract …

Line 104 – for HPLC – please give a name/brand, company, country

Lines 318 – 329 - this section of C. papaya description should go to the Introduction 

Author Response

Response to Reviewer 2

Comments and Suggestions for Authors

In this manuscript authors described the safety profile of C. papaya leaf ethanolic extract in Sprague-Dawley rats. The authors performed the chemical analysis of extract and acute and chronic toxicity studies with different dosses of the extract in rats. The acute toxicity test was performed by using a single oral dose of 5000 mg/kg body weight and the chronic toxicity study by using daily oral doses of 100, 400, 1000, and 5,000 mg/kg over 180 days. Final results of the tests have shown that C. papaya leaf extract exhibited no adverse effects on the rats during these studies. This study is important because it gives a new data on the safety profile of C. papaya leaf ethanolic extract with higher doses. The tests are well designed and executed, and the obtained data is presented in a clear and coherent manner. The methods applied are adequately described. The final concluding remarks are appropriate and supported by the data presented in the manuscript. The references cited are relevant and sufficient. Tables and figures are well prepared. The manuscript is well written and easy to read even though the manuscript needs some additions/corrections (described below):

Line 2 – in the title it should also stay Leaf  … Carica papaya L. Leaf ethanolic extract …

Response: Thank you very much for pointing this out. In order to enhance clarification, we agreed that the term "leaf" was appended to the title; the singular was selected to emphasize the papaya leaf's meaning.

Line 104 – for HPLC – please give a name/brand, company, country.

Response: Thank you very much for pointing this out. The following information has been added to topic 2.3: name/brand, company, and country for HPLC.

Lines 318 – 329 - this section of C. papaya description should go to the Introduction.

Response: Thank you very much for pointing this out. We rearranged this section such that it serves as the introduction.

Reviewer 3 Report

Comments and Suggestions for Authors

The manuscript brings important information about a plant drug (Carica papaya leaves), bringing enough details (in Material and Methods and Results) to a good discussion about acute and long term use of that herbal medicine. 

As the evaluation focused on female rats, the effects on pregnancy and fetuses, in my opinion, will make the text more complete.

Comments on the Quality of English Language

Some typos and grammatical mistakes were observed.

Author Response

Response to Reviewer 3

Comments and Suggestions for Authors

The manuscript brings important information about a plant drug (Carica papaya leaves), bringing enough details (in Material and Methods and Results) to a good discussion about acute and long-term use of that herbal medicine. As the evaluation focused on female rats, the effects on pregnancy and fetuses, in my opinion, will make the text more complete.

Response: Thank you very much for pointing this out. The objective of our study is currently focused on the registration of herbal products that do not necessitate such testing including, the effects on pregnancy and fetuses. Thus, reproductive studies are unnecessary for the papaya leaf extract product to be registered as a health herbal product in accordance with the Thai FDA's criteria. However, if it is subsequently developed and registered as a pharmaceutical product in accordance with the Thai FDA's criteria, we are obligated to adhere to the individual who may do so (pregnancy and fetuses). Additional research is necessary to clarify the implications for the embryo and reproductive system, as well as to investigate the pharmacokinetics and pharmacodynamics. This response will not be included in our manuscript.

Reviewer 4 Report

Comments and Suggestions for Authors

The design of this study is reasonable and the data is rich. The manuscript was well organized and written. This study has a certain significance and value and can be published in “Toxics” after the following revisions and responses:

1. Line 2: According to the content of this study, the “Carica papaya Ethanolic Extract” in the Title should be changed to “Carica papaya Leaves Ethanolic Extract”.

2. Line 79: Why did the authors choose 3 to 4 month old plants?

3. Line 94: Why did the authors use 10% ethanol for extraction? We usually think that the ethanol extract refers to substances extracted with absolute ethanol. The substances extracted with 10% ethanol are more like aqueous extracts. Thus, what is the basis for the authors to choose 10% ethanol here?

4. Table 2:For microbial contamination testing, it would be better if the total number of bacterial colonies and the number of mold colonies could be provided.

5. In Table 13 and 14, to make the table self-explanatory, the full name of the acronyms, such as BUN, SGOT, SGPT, et.al, should be provided as footnote below the tables. In addition, the full name of the acronyms also need be provided when they first appear in the text, such as Line 452.

6. Line 492: uterus in the male satellite group? Do male rats have a uterus?

7. The conclusion section should list the limitations of this study.

8. All the available URLs of the References should be provided.

Author Response

Response to Reviewer 4

Comments and Suggestions for Authors

The design of this study is reasonable, and the data is rich. The manuscript was well organized and written. This study has a certain significance and value and can be published in “Toxics” after the following revisions and responses:

  1. Line 2: According to the content of this study, the “Carica papaya Ethanolic Extract” in the Title should be changed to “Carica papaya Leaves Ethanolic Extract”.

 Response: Thank you very much for pointing this out. In order to enhance clarification, we agreed that the term "leaf" was appended to the title; the singular was selected to emphasize the papaya leaf's meaning.

  1. Line 79: Why did the authors choose 3 to 4 month old plants?

Response: According to an unpublished study by the project group, the papaya leaves were harvested since cultivation in several age of plant. Nevertheless, the leaves were specifically chosen for harvest at the 3–4-month-old stage, because the plants contained the highest concentrations of vital nutrients and bioactive compounds.

  1. Line 94: Why did the authors use 10% ethanol for extraction? We usually think that the ethanol extract refers to substances extracted with absolute ethanol. The substances extracted with 10% ethanol are more like aqueous extracts. Thus, what is the basis for the authors to choose 10% ethanol here?

Response: Thank you for your suggestion. We all agreed that the extract should be water-based. We insist on the phrase "extract" referring to 10% ethanol extract. As a result, we modified the word "extract" throughout the manuscript to "10% ethanolic extract". We have explained more in this response. Our group discovered in a prior investigation that a 10% ethanol extract exhibited favorable pharmacological effects (unpublished data) in relation to the subject aspect of "Implications for Patients with Thrombocytopenia: Research and Development of Papaya Extract." This research can provide scientific support for the Thai FDA registration of papaya extract safety data. As a result, 10% ethanolic extracts have been used in this investigation in accordance with above-mentioned pharmacological activity tests. The registration requirements of the Thai FDA also state that the efficacy and toxicity tests must use the same extract.

  1. Table 2:For microbial contamination testing, it would be better if the total number of bacterial colonies and the number of mold colonies could be provided.

Response: ​Principles to develop an extract monograph in order to standardize the extract in terms of microbial contamination. This is required to be conducted in a standardized setting, and the results will be reported in accordance with Thailand's FDA procedures and standards. As a result, it will simply be stated whether it was “found” or “not found”, and whether it was found or detected above the standard levels or not. As a result, it was not reported as reviewer indicated.

  1. In Table 13 and 14, to make the table self-explanatory, the full name of the acronyms, such as BUN, SGOT, SGPT, et.al, should be provided as footnote below the tables. In addition, the full name of the acronyms also need be provided when they first appear in the text, such as Line 452.

Response: Thank you very much for pointing this out. We have changed the name on the table as you suggested.

  1. Line 492: uterus in the male satellite group? Do male rats have a uterus?

Response: Thank you very much for pointing this out. We wrote the wrong information. This sentence has been edited to be completely correct.

  1. The conclusion section should list the limitations of this study.

Response: Due to the ethical concerns associated with animal testing in laboratories, strict compliance to protocols is required throughout the experimental procedure. Additionally, laboratory animal care and research must be conducted in a standardized setting. We have added this limitation into the conclusion sector.

  1. All the available URLs of the References should be provided. 

Response: Thank you very much for pointing this out. We have attempted to include every URL that is feasible and have carefully reviewed all references.

Reviewer 5 Report

Comments and Suggestions for Authors

 1.    In the methods part, the authors stated that the acute and chronic toxicity evaluation was performed in accordance with OECD Test Guideline 420 and 452 (see lines 134, 145, 429 etc.). However, this is not true. There are several discrepancies between the present study design and the corresponding OECD Test Guideline, e. g. regarding the number of tested rats per group (10 in this study (see l. 145 – 154), 20 animals are needed following OECD GL 452 (see page 5, second paragraph). The same applies in the context of the administered top dose (I the study 5000mg/kg were used but in OECD GL 452 it is mentioned that “The top dose should not exceed 1000 mg/kg body weight/day”. There are further discrepancies between this study and the design proposed in the corresponding OECD Guidelines. Thus, the study should be repeated in a more adequate design to fulfil the OECD GL 420 and 452 and to provide more meaningful results in the context of risk assessment of C. papaya leaf extract to mammals (including humans) and to consumers in general.

2.    “Material and Methods”: Please include also the year of harvest in point “2.1. Plant Material”, line 80. Moreover, in parts describing the toxicity studies (2.6 and 2.7) please also include information how test substance and control were administered (water, feed or whatever). In this context, information of food / water consumption (containing test substance) by rats is missing. This is needed to allow an evaluation of the potential impact of test substance uptake by rats on the measured biological processes/effects (see l. 155 ff).

3.    “Material and Methods”: Please include complete provider information, including city, state, and country for all methods, materials and equipment used in this study (see lines (e.g., see “HPLC” in line 104; also see l. 110, 136 etc.). Sometimes this information is completely missing and have to be included (also see “author guidelines”).

4.    “Material and Methods”: In the part dealing with statistics (l. 162 ff), the authors stated that data were analysed via t-test and one-way ANOVA. However, before using this normal distribution have to be checked. If it is normal distributed this would be ok. However, in rodent studies it is rather unexpected or unclear whether there is a Gaussian distribution (especially when data were obtained from the same animal at different time points (e. g. body weight etc.). Tus, statistical analyses should be re-evaluated by an independent institution or service provider.

5.    Results: Beside absolute organ weight changes (see table 6) also relative changes of tissue weight (e. g. heart vs brain weight and so on) should be evaluated to allow a mor meaningful analysis of the obtained data.

6.    Results: Please include also data obtained for acute toxicity in male animals in the study as done for females in table 5 and 6. Generally, all data relevant for the data evaluation and conclusions made by the authors have to be included instead of only subsets of data. This will allow to follow and evaluate the author statements made in the text.

7.    Tables 11-14: In the collum “organs” there were items, such as RBC, basophils etc. These items are on organs. Thus, please revise the table design and delete the term “organs” in this position. Please use thousands separators or not in numbers, such as 1000 or 5000 vs “1,000” and “5,000” (e. g. see table 11, l 288 vs text l. 284; check the whole manuscript). Use “SEM” instead of “S.E.M.” in the legends of each table.

8.    References: Please check the references (e. g. regarding the use of full vs abbrev. journal names, the use of doi numbers etc.) and use a uniform style for all included references.

Comments on the Quality of English Language

Grammar, style and phrasing (English style and editing): Sometimes space characters are missing or too much in the text (e.g., see line 157 “day180” etc.; check the whole manuscript). Use a uniform style for words including the term “anti” with or without hyphen (e. g. see l. 47 “anti-inflammatory” vs l. 49 “antibacterial [7], antimalarial” and so on. Use a uniform style for units (use abbreviated terms or full names, e. g., compare “kilograms” vs “kg” vs “grams” in lines 97 vs 133, l. 117 etc.). Please define all abbreviations used in the manuscript (e. g. in line 108: „i.d.“, l. 301 “ALP”, etc.). Use a uniform style for the term “SEM” (l. 169) vs “S.E.M.” (l. 249). Please use thousands separators or not in numbers, such as 1000 or 5000 vs “1,000” and “5,000” (e. g. see table 11, l 288 vs text l. 284; check the whole manuscript).

Author Response

Response to Reviewer 5

Comments and Suggestions for Authors

  1. In the methods part, the authors stated that the acute and chronic toxicity evaluation was performed in accordance with OECD Test Guideline 420 and 452 (see lines 134, 145, 429 etc.). However, this is not true. There are several discrepancies between the present study design and the corresponding OECD Test Guideline, e. g. regarding the number of tested rats per group (10 in this study (see l. 145 – 154), 20 animals are needed following OECD GL 452 (see page 5, second paragraph). The same applies in the context of the administered top dose (I the study 5000 mg/kg were used but in OECD GL 452 it is mentioned that “The top dose should not exceed 1000 mg/kg body weight/day”. There are further discrepancies between this study and the design proposed in the corresponding OECD Guidelines. Thus, the study should be repeated in a more adequate design to fulfil the OECD GL 420 and 452 and to provide more meaningful results in the context of risk assessment of C. papaya leaf extract to mammals (including humans) and to consumers in general.

Response: We sincerely appreciate your guidance and wish to extend our sincere apologies for our error. We design and implement all animal experiments in accordance with WHO and OECD guidance for our research. We neglected to include the WHO guideline in our reference. Thus, we have added WHO guideline in our manuscript. Furthermore, we aim to provide further elucidation and supplementary details as follows:

  1. Our objective is to register papaya leaf extract, which is an herbal product, in compliance with Thai FDA regulations. In this experiment, we carried out it in accordance with the following sets of guidelines: WHO and OECD guidelines. The WHO guideline will cover herbal medicine research and testing, whereas the OECD guideline will cover research and testing of compounds with known structural formulae (chemicals). Because this is a test of crude herbal extracts, the WHO guideline will be applied as the principal basis, with the OECD guideline serving as a support, including the details of methodology and procedure for safety testing.
  2. Regarding the design of animal number in chronic toxicity testing, the WHO guideline specifies 10 animals/group/sex, whereas the OECD guideline recommends 20 animals/group/sex. In accordance with the following instructions, especially OECD guideline, the animals in each group were to be divided into the following groups: treated group, interim group, and sentinel group. Due to the numerous factors that influence the safety of test substances, including animal welfare management, animal health, and test substances for the purpose of obtaining accurate safety information, 20 experimental animals should be divided into three groups: a treated group of 10 animals/group/sex, an interim group of 5 animals/group/sex, and sentinel 5 animal/group/sex. In accordance with both guidelines, the number of experimental animals used for statistical analysis must be adequate. In our study, this research complied with both guidelines; therefore, a total of ten animals/group/sex were used in the treated group for experimental.
  3. Furthermore, ethical committees and their consideration must be incorporated into this research. When evaluating projects, the committee will primarily apply the 3Rs (replacement, refinement, and reduction) principle. Indeed, 10 animals/group/sex were deemed an excessive quantity by the committee for the treated group because 5-6 animals/group/sex were adequate for statistical analysis. To comply with both guidelines, we stated that to committee should follow the guideline. Thus, 10 animals/group/sex were granted permission for use by the committee for the project's proposal.
  4. Previous our research on the pharmacological effects of the animal with thrombocytopenia has established that effective doses ranging from 400 to 1,000 mg/kg of extract. As a result, a part of the aim of this experiment is to determine whether or not the extract can be administered to humans. Thus, in order to determine the maximal dose of the test substance, it does not induce toxicity. At this stage of the experiment, the dosage must specify the spacing between the pharmacological dose and the maximum non-toxic dose.

In our study (the OECD 420 for acute toxicity), it is indicated that 5,000 mg/kg is applied without inducing toxicity. This procedure of testing is contained in Annex 4 (page 13/14), paragraphs 19 and 24 (page 4/14), and paragraph 8 (page 2/14). Thus, the maximum dose evaluated for chronic toxicity in experimental animals was chosen to be 5,000 mg/kg.

In accordance with OECD 452, a maximal dose of 5,000 mg/kg was evaluated for chronic toxicity which is mentioned in paragraph 24 (page 6) - “The dose level spacing selected will depend on the characteristics of the test chemical, and cannot be prescribed in this Guideline, but two to four fold intervals frequently provide good test performance when used for setting the descending dose levels and addition of a fourth test group is often preferable to using very large intervals (e.g., more than a factor of about 6-10) between dosages. In general the use of factors greater than 10 should be avoided, and should be justified if used”

      Regarding dosing applied for acute toxicity and dosing applied for long-term toxicity, the General Guidelines for Methodologies on Research and Evaluation of Traditional Medicine of the World Health Organization (WHO) do not specify that in general (page 28 for acute toxicity and page 30 for long-term toxicity). A lethal dose of the test substance is required to induce acute toxicity. In order to assess long-term toxicity, the test substance must be administered in three doses, one of which must be non-toxic. One is a dose-response relationship for toxic manifestations, and another is the overt toxic effect.

  1. “Material and Methods”: Please include also the year of harvest in point “2.1. Plant Material”, line 80. Moreover, in parts describing the toxicity studies (2.6 and 2.7) please also include information how test substance and control were administered (water, feed or whatever). In this context, information of food / water consumption (containing test substance) by rats is missing. This is needed to allow an evaluation of the potential impact of test substance uptake by rats on the measured biological processes/effects (see l. 155 ff).

Response: Thank you for your pointing out.

  • We have added the year of harvest in our manuscript as shown in tracking changes.
  • In all animal studies, the test substances (extracts) were administered to animals using oral gavage without combining with water or food. Moreover, our test design follows both WHO and OECD guidelines. In WHO guideline (General Guidelines for Methodologies on Research and Evaluation of Traditional Medicine) on page 30, states that "For all experimental animals, the general signs should be observed daily and body weight and food intake should be measured periodically. If useful, water intake should also be determined." and OECD guideline 452, pages 14, it is noted that water and food consumption can be reported as if applicable. And throughout the experiment, the animals' health, as well as their water and food consumption, were monitored on every day, and all results were found to be normal. Thus, the results of food and water consumption were not reported in our manuscript.

  1. “Material and Methods”: Please include complete provider information, including city, state, and country for all methods, materials and equipment used in this study (see lines (e.g., see “HPLC” in line 104; also see l. 110, 136 etc.). Sometimes this information is completely missing and have to be included (also see “author guidelines”).

Response: Thank you very much for pointing this out. We have added topic 2.3 to be Chemicals and Reagents to give more information about some materials, equipment, as well as reagents.

  1. “Material and Methods”: In the part dealing with statistics (l. 162 ff), the authors stated that data were analyzed via t-test and one-way ANOVA. However, before using this normal distribution have to be checked. If it is normal distributed this would be ok. However, in rodent studies it is rather unexpected or unclear whether there is a Gaussian distribution (especially when data were obtained from the same animal at different time points (e. g. body weight etc.). Thus, statistical analyses should be re-evaluated by an independent institution or service provider.

Response: We sincerely apologize for the error and thank you for directing our attention to some statistical research that would have improved the precision of our analysis. In order to assess statistical significance, we applied the Mann-Whitney test for comparisons between two independent groups and the Kruskal–Wallis one-way analysis of variance test for comparisons involving more than two groups. Upon careful examination, we determined that ANOVA tests show much more power than Kruskal-Wallis. Nevertheless, the outcomes exhibited a consistent trend, with the exception of the lymphocyte count in female rats administered 1,000 mg/kg, which did not to differ significantly. Hence, within the "Material and Methods" section, the Mann-Whitney test is used for the comparison of independent groups, while the Kruskal-Wallis’s test is applied for the comparison of more than two groups.

  1. Results: Beside absolute organ weight changes (see table 6) also relative changes of tissue weight (e. g. heart vs brain weight and so on) should be evaluated to allow a mor meaningful analysis of the obtained data.

Response: Thank you for your suggestion. The experimental results in Tables 7 and 8 show that the body weights of female and male rats were not significantly different. As a result, there was no measurement for relative organ weight. Moreover, OECD guideline 452, pages 15, it is noted that organ weight ratio can be reported as if applicable.

  1. Results: Please include also data obtained for acute toxicity in male animals in the study as done for females in table 5 and 6. Generally, all data relevant for the data evaluation and conclusions made by the authors have to be included instead of only subsets of data. This will allow to follow and evaluate the author statements made in the text.

Response: Our test design is based on both guidelines (WHO and OECD). The WHO guideline indicates on page 28 in the section on acute toxicity, "In at least one of the species, males and females should be used." with the following limitation: "In the case of rodents, each group should consist of at least five animals per sex." In addition, the OECD 420guideline state in paragraph 1 that "testing in one sex (usually females) is now considered sufficient." Furthermore, the Animal Ethics Committee agreed that this section should be carried out in accordance with the OECD guideline and with the use of single-gender female experimental animals. As a result, the research team first focused specially on females, who are more sensitive to the substance.

  1. Tables 11-14: In the collum “organs” there were items, such as RBC, basophils etc. These items are on organs. Thus, please revise the table design and delete the term “organs” in this position. Please use thousands separators or not in numbers, such as 1000 or 5000 vs “1,000” and “5,000” (e. g. see table 11, l 288 vs text l. 284; check the whole manuscript). Use “SEM” instead of “S.E.M.” in the legends of each table.

Response: Thank you very much for pointing this out. We have modified those words.

  1. References: Please check the references (e. g. regarding the use of full vs abbrev. journal names, the use of doi numbers etc.) and use a uniform style for all included references.

Response: Thank you very much for pointing this out. We have attempted to include every URL that is feasible and have carefully reviewed all references.

Grammar, style and phrasing (English style and editing): Sometimes space characters are missing or too much in the text (e.g., see line 157 “day180” etc.; check the whole manuscript). Use a uniform style for words including the term “anti” with or without hyphen (e. g. see l. 47 “anti-inflammatory” vs l. 49 “antibacterial [7], antimalarial” and so on. Use a uniform style for units (use abbreviated terms or full names, e. g., compare “kilograms” vs “kg” vs “grams” in lines 97 vs 133, l. 117 etc.). Please define all abbreviations used in the manuscript (e. g. in line 108: „i.d.“, l. 301 “ALP”, etc.). Use a uniform style for the term “SEM” (l. 169) vs “S.E.M.” (l. 249). Please use thousands separators or not in numbers, such as 1000 or 5000 vs “1,000” and “5,000” (e. g. see table 11, l 288 vs text l. 284; check the whole manuscript).

Response: We are apologized for the error and appreciate your assistance in improving the manuscript. Grammar and data issues have already been found and corrected for consistency.

Round 2

Reviewer 5 Report

Comments and Suggestions for Authors

Important reviewer comments were noch adequately addressed, esp. regarding the study design.